# E-Communication of ENGO's for Measurable Improvements for Sustainability

**Valentina Burksiene [1] and Jaroslav Dvorak [1,2,*]** 

[1]   Department of Public Administration and Political Sciences, Faculty of Social Sciences and Humanities, Klaipeda University, H. Manto 84, 92294 Klaipeda, Lithuania; valentina.burksiene@ku.lt
[2]   School of Public Management, University of Johannesburg, Governance & Public Policy, Johannesburg 2006, South Africa
*    Correspondence: jaroslav.dvorak@ku.lt

**Abstract:** Environmental non-governmental organizations (ENGOs) play a significant role in contemporary governance. They act as bottom-up advocates while discussing sustainability and environmental issues. They try to engage different stakeholders and society members for common actions. Communication is cited as a very appropriate process for the cooperation and coordination of joint actions. Digital technologies provide new communication possibilities as an e-communication mode that covers various networks. E-communication is very complex and requires strict management that is usually unaware of for small ENGOs. This study aims to propose a theoretical model of e-communication for enhancing ENGOs communication effectiveness on sustainability issues. A literature analysis was used with a content approach helping to collect components and criteria for the framework. The approach of logical classification and distribution was applied to construct the framework. The framework appeals to the idea of diversification of communication for different audiences and is based on the e-communication objectives and measurement of messages as the results. The framework can be adapted to the particular sustainability problem such as air pollution, protection of trees, etc.

**Keywords:** effective communication; e-communication; communication management; environmental non-governmental organization; sustainability

## 1. Introduction

According to Korten (1990), the power of social movements has evolved for decades from their primary attention, which focused on the immediate and visible needs of people (social movements as doers) to the fourth generation based on a people-centered development vision on a global scale (social movements as activists and educators). Examples of social movements (see in Korten 1990) prove them to be a strong power for changing the quality of social life and sustainable development. Currently, social movements have become more organized and active. As nongovernmental organizations (NGOs) they foster authorities to better serve the public and to protect society's rights (Powell 2013; Palttala et al. 2012) and environment. Thousands of NGOs have connected around the globe to impact policies on (Urry 2015) and proved that unified society groups can raise their voice in the decision process and give effective support for clean policy (D'Amato et al. 2015). ENGOs are valued as very active participants in contemporary society that deal with various environmental issues (Sidor and Abdelhafez 2021; Herranz de la Casa et al. 2018; Kilger et al. 2015). NGOs are primarily seen as social development agencies with an advocacy role, but their achievements depend on the capacity to communicate ideas, missions and goals (Hue 2017).

The concept of communication means a process of interaction between or among people through five basic elements: information sources, receivers, messages, channels, and

feedback (Özmen and Karabatak 2013). This process is oriented to the different audiences in order not only to inform about the serious and important issues or promote ideas. Communication is about the understanding of other parties and the engagement with community members, residents and various stakeholder groups for the development of appropriate messages in order to exchange and share ideas or attitudes.

Various communication have recently become available for communication. Studies propose different communication forms, such as mass media, newsletters, recorded media, school curricula, major media events, study groups and social networks, etc. (Korten 1990; Merry 2012; Palttala et al. 2012; Ruehl and Ingenhoff 2015). According to Hue (2017), Pavlovic et al. (2014), Jönsson et al. (2016), it is very beneficial for social movements to use social media channels for effective and efficient communication. According to Di Tullio et al. (2021, p. 2), social media allows "to set a more dialogical and valuable communication" and is crucial in many communications campaigns (Varmus and Kubina 2016). Advanced technologies foster social movements to create and rapidly develop social networks for their communication (Hue 2017; Pavlovic et al. 2014).

Communication is recognized as crucial when coping with very complex and uncertain sustainability challenges. Communication helps to explore, learn and shape sustainable development (Newig et al. 2013; Burksiene et al. 2018).

Schäfer (2012) examined the use of online communication by stakeholders in articulating climate change and found that ENGOs are champions of such communication. According to Yan et al. (2018), many NGOs turn to socioenvironmental problems. Environmental NGOs (ENGOs), according to the report by Burson Marsteller (see Herranz de la Casa et al. 2018), are treated as very serious agents of change for the sustainable future. In the EU they are recognized as the second most effective type of organization after business associations. We guess that sustainability issues should be on top for ENGOs and thus be antecedent in their communication. Hence, the decision to analyse the communication of ENGOs in the field of sustainability seems to be important and logical when analysing communication management.

If well managed, communication can help the organized social groups not only to inform the populace about existing problems but also to offer innovative ways or joint actions for solving common issues by serving as advocates or advocacy organizations (Dellmuth et al. 2020; Bloodgood et al. 2014; Hue 2017; Weder and Samanta 2021).

As its research question, this study aims to address the question of whether communication management can result in measurable improvements for sustainability.

Scientific approaches of systemic literature analysis, conceptual content analysis, logical classification and reduction were used (see detailed description in Section 4: Materials and Methods).

In this article, we revealed the importance of that actions of ENGOs for a sustainable future. We also agreed that e-communication is a very appropriate tool for sustainability issues. However, we found that e-communication on sustainability issues lacks a more complex analysis from the management perspective. Upon agreeing on the necessity of professional management in the e–communication process for its effectiveness, we developed and proposed a theoretical framework that includes different modes of communication *of*, *about* and *for* sustainability based on a management cycle. We concentrated on two aspects of management in particular: (1) planning and (2) measuring phases that can be named as starting and ending points for the effective implementation of a sustainability vision through e-communication. A practical explication of the framework was sampled on issues relating to air pollution.

## 2. Results

### 2.1. Sustainability Communication Management

Communication helps the communicators to reach specific goals of the communication campaign and to attract major voices of potential promiscuous audiences. Therefore, it is very important to treat the designing of any single message seriously, as messages do not

only affect the attitude and behaviour of the intended audience. They also influence the entire communication strategy (Hue 2017). In other words, the vision, goals and objectives for sustainability need to be perceived similarly and agreed upon by the strategists and expressed in an appropriate message related to the appropriate target audience and location.

Newig et al. (2013) have asserted the diversification of communication. The authors defined three modes of sustainability communication considering the communication of, about and for sustainability. According to the authors, this would allow for the shifting of sustainability discourse from neutral or cognitive discourse (communication of sustainability—CoS) to a normative orientation (communication about—CaS and for sustainability—CfS). We agree with this statement and propose that, in order to be heard, environmental movements (ENGOs) need not simply communicate sustainability issues per se, but address different messages of sustainability (CoS), about sustainability (CaS) and for sustainability (CfS) to different audiences. These messages thus could lead to commonly agreed and contextualized knowledge, better collaboration and networked decisions as everyone would receive the appropriate messages and thereby improve understanding and beliefs in sustainability.

In agreeing with Newig et al.'s (2013) propositions, we can develop and explain the meaning of each communication mode. In some cases, sustainability messages can be simplified in order to express sustainability as cognitive knowledge or discourse with the aim to provide facts and the knowledge *of* sustainability (educational mission). This mode is defined as one-way communication. For those with more understanding interpretations, opinions and debates about sustainability issues could be messaged. These messages can help to conceptualize sustainability issues and and create common understanding of sustainability. This mode of communication is perceived to be two—way horizontally networked and to express not only agreements but also to reveal controversial opinions as well. When the message aims to involve or invite people to participate in common activities it can be designed as exhortation to join for sustainability actions (consolidation mission). It is proposed that all involved stakeholders transform towards the normative sustainability goals when participating the such communication leading to common actions. According to the authors, communication for sustainability may integrate and share other two modes, and with this they prove interrelation existing among all three modes.

These modes (CoS, CaS and CfS) substantiate the importance of different messages to diverse audiences in the ENGOs networking cooperation, and we thus set them as central ones in our theoretical framework of effective communication management in the context of sustainability (Figure 1).

As we analyse communication from the point of ENGOs, we reference a bottom-up communication that can be directed both vertically and horizontally. According to the communication modes by Newig et al. (2013), communication *of* sustainability flows vertically and horizontally in a one-way direction from a sender to a receiver, while communication *about* sustainability is proposed to be horizontal with a two-way flow. This communication can create horizontal networks. When communication *for* sustainability goes into action, we can receive a two-way communication network both horizontally and vertically. This idea should be considered in the implementation of the framework.

Components of the management cycle are logically necessary for effective communication. Studies by Raupp and Hoffjann (2012), Rajhans (2018), Herranz de la Casa et al. (2018) prove that communication management is beneficial when the problem cannot be solved by employing ordinary or routine actions. These are sustainability issues that cannot be solved in everyday routine and need effective communication management: planning with the inclusion of vision, mission, and objectives as well as monitoring and measuring the progress of the implementation or its result (Andres 2011; Palttala et al. 2012; Hue 2017; Mihai 2017; Rajhans 2018; Herranz de la Casa et al. 2018; Burksiene and Dvorak 2020). Planning the communication objectives would start the management cycle, and measures would evaluate the success of the implementation of that communication strategy in the end. Traditionally, management is exposed as a cyclic model, but in our framework, the

picture is simplified and the management process is depicted in a linear model at the bottom of a framework (Figure 1).

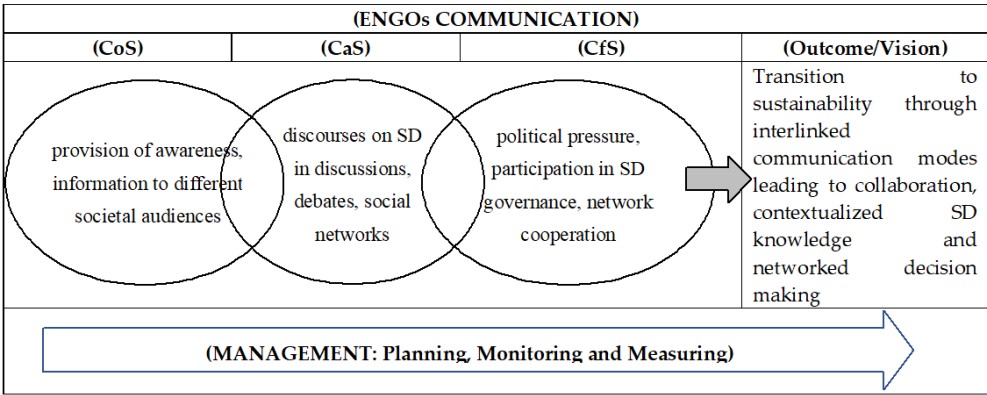

**Figure 1.** ENGOs effective communication management towards sustainability.

Employment of this framework means that in order to reach the outcome (or vision), ENGOs should:

1. integrate all three modes (CoS; CaS and CfS) when planning communication,
2. monitor the plan of communication while implementing,
3. measure the results of communication strategy.

The contents proposed by Newig et al. (2013) (see definitions enshrined in the centre of every communication mode of Figure 1) are useful prompters for the steps to be fulfilled and can be treated as the objectives in a communication strategy (Table 1).

**Table 1.** Objectives for effective communication.

| (Mode) | (Recommended Objectives) |
|---|---|
| **CoS** | 1. To provide awareness and information of sustainable development as per se<br>2. To define targeted societal audiences for communication about sustainable development |
| **CaS** | 3. To agree on sustainable development discourse related to particular territory<br>4. To organize debates about sustainable development with target audiences<br>5. To develop social networking |
| **CfS** | 6. To increase pressure on political decisions<br>7. To participate in governance related to sustainable development<br>8. To improve network cooperation for sustainable development |

Considering the interrelation of all three communication modes, recommended objectives are also considered to be interrelated. These communication management objectives logically start from a concentration on a common understanding of sustainability by the target audiences and lead to the increased actions for sustainable development via improved network cooperation.

It is pivotal that sustainable development is proposed to be locally adapted (Zahra et al. 2011; Bell and Morse 2003), therefore activities for the realizing of every objective are recommended to be developed considering specific sustainability issues and unique societal environment on the particular location.

Communication actually needs systematic monitoring (periodic assessment of the progress of planned measures) and is based on communication measurement indicators. Monitoring or control of communication can become more integrated by involving multiple devices and the Internet of Things (Nordin et al. 2021). Accordingly, monitoring of activities (as an integrated part of a management cycle) depends on decisions and projections at the local level and requires a unique individualized design. The design of a monitoring group

is directly related to the particular organization and engaged stakeholders with the main role dedicated to the leader of the organization (Alolabi et al. 2021). The leader is perceived as a primary initiator of the entire communication management who should enhance the organization towards this complex process and convince all members of the overall value of the innovative communication.

*2.2. Measurement of E-Communication*

Nongovernmental organizations (NGOs) can help to strengthen society's power in a bottom-up way. They strategically focus on public policy problems, the implementation of activities, and cooperation areas. NGOs play a significant role (advocacy, representation and mobilization) while engaging various actors in a social change process, which also implies different communication strategies (Bies et al. 2013; Yan et al. 2018). These organizations can achieve positive changes in many global and local issues.

Communication is proposed to be a very beneficial instrument (Herranz de la Casa et al. 2018; Mihai 2017; Muszynska 2017; Liu 2012) that helps to create networks connecting society members, businesses and governments both vertically and horizontally. From the point of view of NGOs, horizontal communication can address a positive reshaping of social movements driven by ideas and a vision of a better world (Korten 1990), and in a vertical manner it can encourage governments to listen to the requirements or needs of citizen groups. Thus, the horizontal and vertical networks of different stakeholders prove that communication is complex. Polizzi and Murdie (2019) have described how local NGO communication with international NGOs creates opportunities for government pressure. According to them, local NGOs are sending a "boomerang" message to international NGOs. International NGOs are sending a "boomerang" message back by promising logistical and material assistance to local NGOs and putting pressure on the government.

By agreeing on two modes of performance management (Kroll et al. 2019; Andres 2011), it is perceived that NGOs communication, as a *bottom-up* communication, can be (i) professionally and constructively managed and administered using modern management principles, or (ii) can exist unmanaged in the medium of disseminating a variety of individual opinions, often unsubstantiated. In the second case, the ongoing communication would not be of great interest to the city/municipal government and other stakeholders or developers (business, education, culture, etc.). In the first case, on the contrary, the city authorities and those responsible for the city's well-being should respond promptly to bottom-up statements, comments, requests, etc. Such communication can attract mass media's attention as well.

In order to be effective, such communication should be managed professionally. However, the idea of professionalising NGOs runs counter to their values, according to a study by Lundasen (2012) on the opposition of Swedish NGOs to EU VAT regulation. Small NGOs, however, work on a voluntary basis and do not give much attention to professional communication practices (Muszynska 2017). Their communicators spend much more time as technicians than as communication strategists (Liu 2012). NGOs lack and compete for different resources such as staff (volunteers, members of the board), finances (sacrifice, membership fee), time, knowledge, etc. Lack of communication management knowledge leads NGOs to unprofessional communication practices and thus to being unheard. According to MacIndoe (2014), the unprofessionalism of NGOs manifests itself through two strategies: specialization and generalization. In the case of specialization, NGO communication focuses on a narrow niche of resources and serves a narrow market, providing a small range of goods and services. Generalists focus their communication on a wider niche, serving a wider market, and offer a wide range of goods and services.

Progressive and innovative digital technologies add much effectiveness to communication, but ENGOs should be creative and skilful when using these non-traditional methods for solving problems (i.e., social media platforms, the internet, blogs). Today, e-communication has become the most preferred type of communication (Calabrese et al. 2021; Özmen and Karabatak 2013) due to its many advantages: place, time, effort, etc. For in-

stance, social media provides the ability to hashtag essential meanings and tag specific users what is necessary to do for attracting more potential audiences and develop networked cooperation. Calabrese et al. (2021) argue that digital platforms enable all actors to channel their efforts towards achieving sustainability goals, but specific skills among different actors and interests are required for effective coordination of this nonlinear process.

Following the theory of communication management, measurement of e-communication by specific criteria should be pursued in order to monitor and evaluate the effectiveness of a communication strategy.

Comfort and Hester (2019) proposed that the effectiveness of the digital communication of ENGOs in social networks can be measured by three dimensions volume, topic (valence), and participants (Table 2). The proposed categories and subcategories help to evaluate messages (frequentness, lengths of stay, shares, etc.) and attitudes, actions or reactions of targeted audiences (support, opposing, responses, etc.) or in other words, to measure them. These subcategories in our study were compared with Newig et al.'s (2013) assessment criteria for Cos, CaS and CfS modes, which originally were not addressed to e-communication (Table 2).

**Table 2.** Dimensions of measurement ENGO's communication effectiveness.

| (Category) | (Subcategories by Comfort and Hester 2019) | (Assessment Criteria by Newig et al. 2013) |
|---|---|---|
| 1. Volume | 1.1. Numbers of sharing of NGOs messages by all users (calculation of shares of every message; more shares more volume) | 1.1. **CaS**: measure the extent to which the discourse in one subsystem (e.g., science) is compatible with discourses in other subsystems (e.g., the political system), and how likely it is to transfer important aspects from one subsystem to another |
| 2. Topic/valence | 2.1. Days for the message supporting NGOs position staying on top | |
| | 2.2. Number of messages with counter-public opinions for a particular message (these messages are not under the NGOs control and can upend its goals) | 2.2. **CfS**: neglect or even obstruct sustainable development symbolically subscribing to sustainability while pursuing hidden non-sustainable agendas |
| 3. Participants | 3.1. Numbers of sympathetic audience of NGO (NGO responds to them and invites actions) | 3.1. **CfS**: to facilitate societal transformation towards the normative goals of sustainable development. |
| | 3.2. Numbers of supportive news media attracted (reactions or shares by news media on the messages of the NGO) | 3.2. **CaS**: the amount of attention that an issue receives from the mass media |
| | 3.3. Who is posting NGOs' content, | 3.3. **CaS**: who has access to the discourse and influences the framing processes; structural conditions and the design of communication processes |
| | 3.4. Who are the target audiences? (Numbers of direct tags and who is "tagged" in the post) | 3.4. **CoS**: Have the recipients been reached? Have they understood the message? Have they, perhaps, changed their values and behavior? |

The results show that five of the six subcategories listed by Comfort and Hester (2019) have logical equivalency with assessment criteria by Newig et al. (2013). Hence, we assert that these five subcategories can help to calculate data and are appropriate in measuring the effectiveness of e-communication management. The criteria represent all three communication modes: CoS, CaS and CfS, and therefore overall communication can be both planned and measured. Measurement of communication finalizes the framework for effective e–communication management.

### 2.3. Sample of Implementation of the Framework Based on Air Pollution Issues

Communication does not happen in a vacuum; it always has content (it is always about something). Issues with regards to sustainability (climate change and other environmental problems in particular) have seen a rapid explosion in the time of the COVID-19 pandemic.

According to Schäfer (2012), ENGOs: (1) provide information on different topics, activities, etc.; (2) seek access to news media; (3) seek to increase the number of supporters through online communication; and (4) change behaviours and mobilize as they can engage supporters in real action. Similar insights are observed in the theory of action (see in Korten 1990), in which the focus is on a vision that needs to be communicated to reach public consciousness and mobilize voluntary actions. The same position is taken by Tumulyte (2012), who states that ENGOs propose new untraditional ways for problem solutions and help to form legal relations between civil society and government. Therefore, we decided to concentrate on the digital communication or e–communication of ENGOs in this study.

As mentioned earlier, in order to be heard, these ENGOs need to focus on different target audiences and send the right messages to the right recipients for communication of the sustainability mission. The audience, according to Burksiene (2016), and Armstrong (2015) variates by a foundational understanding of this complex phenomenon. Therefore, the design of the messages should vary considering the knowledge of the particular audience. This knowledge does not reside only in the mind, but also is situated in the context of past experiences, values, and beliefs, and is influenced by the social environment (Armstrong 2011).

E-communication management is appropriate for any sustainability issue and objectives can be adapted considering local sustainability problems. As an example, the framework is based on air pollution issues. Air pollution is a very serious issue in the context of sustainability. Air pollution in industrial cities has various sources with the inclusion of all industries, transport means and individual housekeepers using unsustainable heating equipment. The literature (Fanø 2019; Marín et al. 2017; Hricko 2012; Burskyte et al. 2011; Belous and Gulbinskas 2008; Sharma 2006; Bailey and Solomon 2004) assumes specific environmental and air pollution issues. Objectives five and eight from Table 1 are linked, as they are very interrelated (Table 3).

**Table 3.** E-communication management against air pollution.

| (Interrelated Communication Modes/Directions) | (Recommended Objectives) | (Measurement Criteria) |
|---|---|---|
| **CoS/vertical and horizontal**<br><br>**CaS/horizontal**<br><br>**CfS/vertical and horizontal** | 1. To provide awareness and information of issues of air pollution as per se<br>2. To define targeted societal audiences for communication of harm of air pollution for sustainable development<br>3. To agree on issues against air pollution discourse related to particular territory<br>4. To organize debates about air pollution with target audiences<br>5. To develop social networking and improve network cooperation against air pollution<br>6. To increase pressure on political decisions<br>7. To participate in governance related to sustainable development | 1.1. Numbers of sharing of NGOs messages<br>2.1. Days for the supporting message staying on top<br>2.2. Number of messages with counter public opinions for a particular message<br>3.1. Numbers of sympathetic audience of ENGO (NGO's responses and invitations for actions)<br>3.2. Numbers of supportive news media attracted<br>3.3. Who is posting NGO content<br>3.4 Who are target audiences (numbers of direct tags and who is "tagged" in the post) |

Communication of air pollution issues should go first in order to increase volume (objective 1). Measurement of numbers with regard to sharing NGOs messages (measurement 1.1) and days for the supporting message staying on top (meas. 2.1) can reveal if the issue is important for the audience and can help to define (categorise) targeted social audiences (objective 2). The audience can be categorized by measuring the numbers of a sympathetic audience of ENGO (meas. 3.1) and those with counter-public opinions for a

particular message (meas. 2.2). These from the 3.1 measurements could be engaged in the implementation of objective 3, agreeing on issues against air pollution discourse related to the particular territory when communicating about air pollution problems and what could be done for achieving better air conditions. Counter public (meas. 2.2) on contrary should be invited to debates about air pollution's impact on the welfare of the local population (objective 4). The outcomes of objectives 3 and 4 would logically lead to objective 5: the ENGO should be ready to develop social networking and improve cooperation against air pollution. Supposedly active social networks would attract news media or, contrary to this, the networking stakeholders would bring their relations with supportive news media (measurement 3.2). Such a social network with its topical issue on air pollution is considered to be ready to increase pressure on political decisions (objective 6) and seek real participation in the governance bodies related to air pollution issues (objective 7). For the improvement of e-communication management, constant measurements of who is posting the content (meas. 3.3) and the target audiences (meas. 3.4) must be implemented.

### 3. Discussion

Sustainable development is the primary problem from which the global idea of sustainable development has emerged. Nevertheless, the problem is recognized to depend not only on global but on local decisions and actions as well. Following best practices, stakeholders must be engaged in the development of programs or action plans concerning issues of sustainability. Sustainable development should be framed in the mode of governance that would expand both the assessment of impact and decision-making processes while engaging not only city authorities but also the city's stakeholders from public, private, academic and voluntary (nongovernmental) sectors.

Two-way symmetrical communication on digital platforms such as social media is a contemporary e-channel that can help active society members to achieve support from the public due to more performance transparency and increased networked cooperation. Calabrese et al. (2021) name all cooperating actors as a meta-organization that develops a digital ecosystem. The success of such an organization mostly depends on the collaborative governance of all involved parties and on clear rules and instructions provided by the platform sponsor. The authors assert that stringent instructions shorten the freedom of discretion and decisions of individual actors of the ecosystem. Actually, ENGOs could hardly be sponsors of digital platforms by themselves as maintenance of this tool requires specific skills and significant finances.

Digital networks can also play an educational role where everyone can transfer information and share knowledge. The authors assert that this tool is not effectively used for the consolidation of society for positive changes. One-way communication is dominant. Di Tullio et al. (2021) researched and proposed sustainability reporting as a beneficial tool for dialogic communication. In the example of Italian universities, the authors assert that announcements of an organization's sustainability report on social media and invite stakeholders to the presentation of these reports to both engage and educate stakeholders and therefore create two–way communication. What is difficult in the implementation of this idea in ENGO is that the organization should be large enough for this initiative. In addition, the ENGO must constantly prepare reports and organize presentations. This costs money, which is usually lacking in all NGOs. Therefore, more research on the topic would help to reveal both restrictions and possibilities to create and successfully develop two-way communication exactly on the example of ENGOs.

NGOs are treated as very powerful participants in local development. They can construct bottom-up two-way relations vertically with the local government and horizontally with community members and different stakeholders. ENGOs are very active participants in contemporary democratic society. They advocate a participatory approach to managing environmental issues towards a sustainable city and global environmental governance. The power of ENGOs is dependent on their innovative provision of activities and trust in technology solutions. ENGOs should not only inform but also educate community mem-

bers with regard to changing their behaviour towards more sustainability. Professionally managed e-communication can empower the voice of ENGOs in cooperating networks. The articles prove that small ENGOs need more financial support and time to educate. However, both financially rich and poor ENGOs seek to educate their audiences. ENGOs serve as an intermediary that conveys scientific findings on sustainability into a political issue that is understandable to the public (Merry 2012). Most members act on a voluntary basis and have scarce resources. In contrast to the large ENGOs, their actions are more fragmented. But it also might be that small ENGOs have advantages as well. Studies comparing differences between the effectiveness of small and large ENGOs could help fill the gap for improvement by benchmarking or by other approaches.

The employment of e-communication management results in higher volume, stronger valence and support of messages, as well as more engaged participants in the two-way vertical and horizontal network aiming to change attitudes of all that are engaged to sustainability issues. Measurement of the communication process is a must for the improvement of e-communication.

ENGOs need to gain knowledge of e-communication management and follow the steps of the management cycle consistently. They should agree on the strategic mission, vision and priorities and clearly articulate them to the audience using all possible communication tools (personal web pages, newsletters, etc.). Indeed, more articles analysing peculiarities of every digital technology for e-communication are necessary.

## 4. Materials and Methods

Sustainability issues, more or less being discussed among politicians and academics for many decades, rapidly exploded during the time of the COVID-19 pandemic. The boost from new technological tools has started changes in all organizations with no exception of ENGOs. The role of these nongovernmental organizations in issues of climate change and environmental protection has been conceded both on political and academic levels (see authors cited in this article above).

Communication is proposed as being very beneficial for speaking out about environmental issues from the bottom-up (Hue 2017). Most authors agree that communication of ENGOs is based strategically on a vision, mission and objectives. There are a few models of communication management proposed in several studies (i.e., Rajhans 2018; Hue 2017; Newig et al. 2013). The aim of these models is to propose how to make an impact on the stakeholders to whom the communication messages are directed, and to try to change their behaviour.

Various aspects of communication have been discussed except for the modelling of effective ENGOs' e-communication management towards more sustainability with the inclusion of planning and measurement of the effectiveness of such communication. This study provides a broader and more complex understanding of e-communication management of ENGOs with orientation to different message content for different audiences that could lead to more effective communication and the realization of the sustainability vision. This study contributes to the consolidation of four main aspects:

(1) the importance and role of ENGOs for sustainability;
(2) the benefits of e-communication in times of digitalization;
(3) ENGOs' e-communication planning and measurement;
(4) the differentiation of stakeholders and society groups considering different levels of their sustainability knowledge and understanding.

No previous studies have adopted the combination of all these theories.

A systematic literature review was used for revealing these aspects. We analysed specific articles on the topics, as there are no studies combining all four discourses.

In the first subsection, we proposed the framework of effective communication management with the inclusion of strategic vision and objectives for a different audience. In the second subsection, we concentrated on revealing the benefit of social media platforms for

e-communication and proposed criteria for measuring the effectiveness of e-communication by ENGOs.

The aim of the article is to clarify related, adequate and optimal models useful for environmental non-governmental organizations to effectively communicate sustainability issues on social media platforms. In order to reach this aim, the approach of logical classification and distribution was applied.

This research follows a logical scientific design and uses appropriate approaches to reveal the answer to the problem question (Figure 2). Theses proposed in this article emphasize the exceptional benefit of e-communication management for ENGOs as bottom–up advocacy if they desire to be heard both horizontally and vertically.

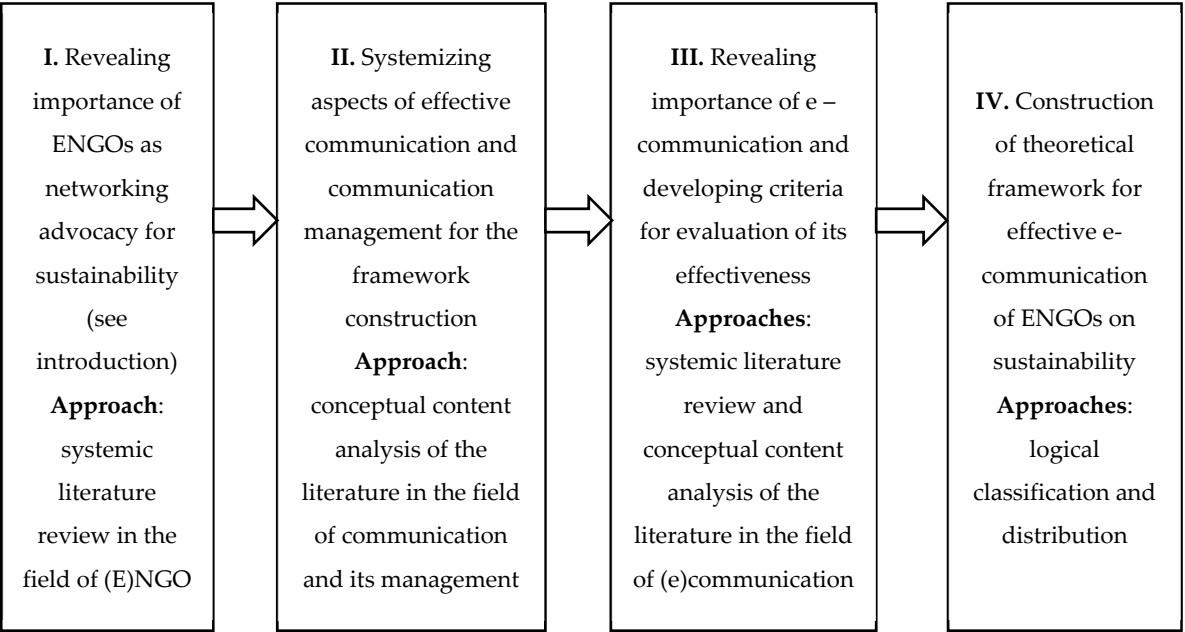

**Figure 2.** Research design.

## 5. Conclusions

The conclusions of this article are in two parts. The first is a contribution to an existing theory. Second, these are the limitations of the current study.

### 5.1. Contribution to Theory

Many articles fragmentally analyse communication management, but a broader view is still missing. This study provides new insights on e-communication management. The framework was adopted from management theory with the integration of the theory on communication of, about and for sustainability. An interdisciplinary approach with a combination of management and communication theories and deep insights could further develop the overall picture.

This conclusion supports some of the assumptions that to be leaders in sustainability issues, ENGOs need to gain knowledge of e-communication management and follow the steps of the management cycle consistently. They should agree on the strategic mission, vision and priorities and clearly articulate them to the audience using all possible communication tools (personal web pages, newsletters, etc.), and not only social networks. Employment of e-communication management results in higher volume, stronger valence and support of messages as well as more engaged participants in the two-way vertical and horizontal network, aiming to change the attitudes of all engaged to pollutant emissions, climate negotiations and regulatory interventions.

### 5.2. Limitations

The framework of effective communication could be useful to research the communication practice of ENGOs and propose improvements enhancing the volume of being heard for sustainability purposes.

We do not insist on our conclusions as being final. The framework needs further development and improvement with more communication objectives and measurement criteria analysed. Measurement of communication is connected to the monitoring process. We excluded this phase from the framework considering that monitoring is deeply individualized and depends on leadership as well as engaged stakeholders. Further research on inclusion monitoring in the framework would help to construct the overall picture of communication management.

At present, the framework can be further developed with more emphasis put on the monitoring and conducting phases. The framework can be adapted by every ENGO acting both on sustainability issues in general and on the particular sustainability problem such as air pollution, protection of trees, etc.

The theoretical framework also needs practical testing. Subsequent future studies should examine other factors of management such as organization, implementation and monitoring. These phases are directly related to a particular organization (people) and can reveal specific variations. Further studies could practically test different topics of sustainability communication. As a next step we are going to test this framework on the issue of air pollution in a port city.

**Author Contributions:** Conceptualization, V.B. and J.D.; methodology, V.B.; software, V.B.; validation, V.B. and J.D.; formal analysis, V.B.; investigation, V.B. and J.D.; resources, V.B. and J.D.; data curation, V.B.; writing—original draft preparation, V.B.; writing—review and editing, J.D.; visualization, V.B.; supervision, J.D.; project administration, J.D.; funding acquisition, J.D. All authors have read and agreed to the published version of the manuscript.

**Funding:** This research received no external funding.

**Institutional Review Board Statement:** Not applicable.

**Informed Consent Statement:** Not applicable.

**Data Availability Statement:** Not applicable.

**Conflicts of Interest:** The authors declare no conflict of interest.

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
