# Peer review of "E-Communication of ENGO’s for Measurable Improvements for Sustainability"

_admsci, doi:10.3390/admsci12020070_

Round 1
Reviewer 1 Report
I consider that the theoretical revision of the article needs to be revised, the sources are not very updated, there are many articles on the subject from the year 2020 that have not been taken into account. In the conclusions, it is necessary to incorporate the projections of the study. It must incorporate the possible applications and scope of the study. You must improve the writing, be careful in the punctuations in some paragraphs. The study is novel, it has theoretical coherence. You should improve from the indicated indications.
Author Response
Dear Colleague,
Here find our answers to your comments on our manuscript.
REVIEWER |
REMARKS |
CORRECTIONS |
Reviewer 1 |
the theoretical revision of the article needs to be revised, the sources are not very updated, and there are many articles on the subject from the year 2020 that have not been taken into account |
Proposed articles by reviewer 3 were analysed and added considering the appropriateness to the topic |
In the conclusions, it is necessary to incorporate the projections of the study. It must incorporate the possible applications and scope of the study. |
Incorporated in conclusion |
|
improve the writing, be careful in the punctuations in some paragraphs. |
Checked by a native speaker |
Reviewer 2 Report
The article is interesting, but needs to be improved. First, the proportions between the parts aren't good. The introduction is too long and does not describe the methods and research questions. At the end of this section, the author (s) formulated a hypothesis, but it should be emphasized more. Otherwise, figures 1 and 2 are the same. There should be a description under the tables.
Author Response
Dear Colleague,
Here find our answers to the comments which you provided on our manuscript.
Reviewer 2 |
the proportions between the parts aren't good. The introduction is too long and does not describe the methods and research questions. At the end of this section, the author (s) formulated a hypothesis, but it should be emphasized more. |
The introduction part has been shortened. Methods and research questions are formulated in this part with more emphasis on the problem instead of more emphasis on the hypothesis. |
Otherwise, figures 1 and 2 are the same. |
It is hard to explain how Fig. 1 disappeared from the paper and Fig. 2 was replaced instead. Original Fig. 1 is placed again. |
|
There should be a description under the tables |
Descriptions are added and improved |
Reviewer 3 Report
I have made some suggestions within the manuscript. In summary I think to be comprehensive the articles needs more examples on how this approach to social media can respond to a specific challenge and how the social media can be assessed for meeting goals relative to specific challenges. The reductionist consideration of social media use for sustainability needs to be connected with specific/localized goals and details provided on how progress would be monitored and how the strategy would be adjusted - again focussed upon specific goals.

Author Response
Dear Colleague,
Here find our answers to the comments which you provided to our manuscript.
Reviewer 3 |
LINE 1 The article looks closely at three different modes of communication for sustainability, considering sustainability communication through a reductionist analysis. Perhaps the current paper represents an opportunity to look more in terms of holism regarding the sustainability process. To make the paper more comprehensive, I suggest adding an analysis of where sustainability efforts are now perhaps using the pollution example given and how the three modes of communication might best be used towards restoration and pollution reduction. Consider the question – ‘How a communication strategy will result in measurable improvements for sustainability’. I don’t think that the argument is well supported towards “more sustainability”, more specifically talks about more communication and refers to a “strategy” that should perhaps be better described in the paper. I have added a few current references from Administrative Sciences to suggest where this article might best be expanded somewhat so as to be more comprehensive. |
The paper surely is based on three modes of communication toward sustainability. This was not only defined in the text but also presented in Fig.1 that somewhen and somehow was changed to Fig.2. Original Fig. 1 is returned in this corrected version of the article. The content of this paper is directed to the development of the theoretic framework for effective e-communication management on the issues of sustainability. The example of air pollution (as one of the sustainability issues) has been chosen to reveal possible practical adaptation of the framework in order for the framework to be more comprehensible for the practicians. We are going to examine this framework in the next research that will be based on the air pollution issue and to reveal where sustainability is at a moment and how these three communication modes can foster pollution reduction in a particular municipality. The title of the article was corrected following the remarks of the reviewer Proposed articles were revised and the manuscript was improved with the appropriate theses. Suggestions have been accepted with improvement main part as well as both discussion and conclusions parts accordingly. Corrections in the text were made in order to shift emphasis from the „strategy“ to „communication management“ |
|
LINE 24 I think it would be good to bring in a reference here on per capita giving in one or more countries – and then also pivot to give a brief review on where sustainability is now, how it has changed in the past 30 years since Korten and a consideration of the critical steps forward in terms of restoration for sustainability. |
This paragraph is more about the evolution of social movements in general. A brief review of transformation of social movements with emphasis on sustainability is added. |
|
LINE 129 Perhaps it would be best to illustrate an entire communication strategy – for example regarding an ENGO response to specific pollution challenges. As suggested above, I think the paper needs to work ‘upstream’ in terms of a holism and localized/specific goals for sustainability to complement and integrate with the details of the communication types. |
Not all components of the communication strategy are included in the theoretical model being constructed. Therefore, in the article, we mostly focus not on the entire strategy itself (as a whole), but on the planning and measurement phases as aspects of the beginning and end of the communication. Corrections in the text were made in order to shift emphasis from the term„strategy“ to „communication management“ A detailed sample on air pollution is added (see the text below Table 3) |
|
LINE 185 These can be best be described in detail for one or several communication strategies so as to provide an example(s) towards how to design a comprehensive approach to developing a communication strategy. |
We have first developed the theoretic framework, based on literature reviews. A detailed sample on air pollution was presented after when the theoretic framework was presented and explained (see the text below Table 3) |
|
LINE 190 This section should perhaps include a consideration of how in-person trust building activities can be best be integrated to develop trust building between entrepreneurial ENGOs and groups/individuals. See Guimtrandy doi.org/10.3390/admsci12020047 |
Indeed, the proposal is valuable in further studies on the topic. At a moment we agree with Korten’s (1990; 126) proposition that the surest way to kill the movement is by drowning them in money. Therefore, we do not treat ENGOs as entrepreneurial organizations. The proposed research paper proposes the idea of pitching between entrepreneurs and investors for trust-building. The idea can be researched while practically testing the proposed framework |
|
LINE 195 Considering Alolabi et al. it would perhaps be best to provide an example of how these modes can respond to a specific situation and turn ‘data into information for better decision making’. doi.org/10.3390/admsci11040140 |
This comment was provided next to Table 1. The objectives in that table are converted from every theoretic communication mode in Fig. 1 (the original version). It is difficult to consider the comment because we are not sure if the reviewer saw the original Fig. 1 and whether the comment is related to Table 1. Anyway, we add the possible example in relation to these modes responding to the sample situation. A citation of Alolabi et al. is also inserted in the manuscript. |
|
LINE 243 Perhaps consider how an individual digital platform for pollution in a specific context/location could be designed to a ‘sustainable meta-organizational communication model for sustainability’ see Calabrese doi.org/10.3390/admsci11040119 |
The proposal is partly considered with some ads in the main text. Our manuscript is about communication as a process, therefore, to direct the research to the development of a digital platforms in this paper would go out of the frames of this topic. Perhaps the development of an individual digital platform would be challenging for any NGO. |
|
LINE 260 Perhaps consider how to approach the need for innovative stakeholder engagement through sustainability reporting (see Tullio doi.org/10.3390/admsci11040151) with reference to a specific example as to what needs to be reported, how often and in what detail.
|
The proposed research was very useful and helped to improve the discussion part. |
Round 2
Reviewer 3 Report
Any further effort on relating the work to the holistic approach to sustainability would likely improve the readability and citation use.